# Identification of miRNA Associated with *Trichomonas gallinae* Resistance in Pigeon (*Columba livia*)

**DOI:** 10.3390/ijms242216453

**Published:** 2023-11-17

**Authors:** Xinyi Li, Aixin Ni, Ran Zhang, Yunlei Li, Jingwei Yuan, Yanyan Sun, Jilan Chen, Hui Ma

**Affiliations:** State Key Laboratory of Animal Biotech Breeding, Key Laboratory of Animal (Poultry) Genetics Breeding and Reproduction, Ministry of Agriculture and Rural Affairs, Institute of Animal Science, Chinese Academy of Agricultural Sciences, Beijing 100193, China; xinyili256@163.com (X.L.); naixin951@163.com (A.N.); zhangrancaas@163.com (R.Z.); mailyunlei@163.com (Y.L.); amstrongyuan@163.com (J.Y.); yanyansun2014@163.com (Y.S.); chen.jilan@163.com (J.C.)

**Keywords:** *Trichomonas gallinae*, trichomonosis, miRNA, pigeon, resistance

## Abstract

*Trichomonas gallinae* (*T. gallinae*) has a great influence on the pigeon industry. Pigeons display different resistance abilities to *T. gallinae*, so the study of the molecular mechanism of resistance is necessary in breeding disease resistant lines. MiRNA plays important roles in the immune response, but there are still no reports of miRNA regulating trichomonosis resistance. We used small RNA sequencing technology to characterize miRNA profiles in different groups. *T. gallinae* was nasally inoculated in one day old squabs, and according to the infection status, the groups were divided into control (C), susceptible (S) and tolerant (T) groups. We identified 2429 miRNAs in total, including 1162 known miRNAs and 1267 new miRNAs. In a comparison among the C, S and T groups, the target genes of differentially expressed miRNAs were analyzed via GO and KEGG annotation. The results showed that the target genes were enriched in immune-response-related pathways. This indicated that the differentially expressed miRNAs had a critical influence on *T. gallinae* infection. Novel_miR_741, which could inhibit the expression of *PRKCQ*, was down-regulated in the T group compared to the C group. It was proven that a decreased novel_miR_741 expression would increase the expression of *PRKCQ* and increase the immune response. This study brings new insights into understanding the mechanism of trichomonosis resistance.

## 1. Introduction

Egypt and Greece domesticated wild pigeons 5000 years ago. After years of development, the pigeon, an important economic animal, has also become the fourth most important poultry in China after the chicken, duck and goose. Pigeons could be divided into edible, ornamental and racing pigeons according to their use, and they have contributed to important research in cranial nerves, behavior, skeletal muscle growth and pigeon milk secretion [1,2,3,4]. Trichomonosis of pigeon is a common protozoal disease, commonly known as mouth ulcers, which is a serious hazard to the health and production of pigeons. Trichomonosis is mainly transmitted by the process of feeding crop milks by parent pigeons to squabs and can be transmitted by pigeons during courtship and through drinking water contaminated with *Trichomonas gallinae* (*T. gallinae*) [5]. The disease is predominantly marked by severe ulcerations and yellowish-white coverings of the mouth, throat and internal organs [6]. In squabs, the oral mucosa is damaged, followed by pathological damage to the liver and spleen, resulting in decreased immune function and increased mortality.

MiRNA is a double-stranded, non-coding, small RNA with a length of about 22 nucleotides that is manipulated by a single-stranded precursor by shearing the Dicer enzyme to form a mature miRNA, forming an RNA-induced silencing complex (RISC) that matches the 3′-terminal untranslated region (3′-UTR) of the target mRNAs for regulation [7]. In the study of parasites, miRNAs interact with the host and the parasite to influence processes such as immune regulation and gene regulation in vivo [8]. Many studies have shown that parasitic infections cause alterations in the host miRNA expression profile [9]. A cryptosporidium infection causes changes in the miRNA expression profile in host epithelial cells [10]. Studies have found that Toxoplasma gondii infections increased the levels of host miR-17, miR-18, miR-19, miR-25, miR146a and miR-155, and these miRNAs could modulate the host’s inflammatory factors [11,12]. Regulation of miRNA expression in erythrocytes can affect their resistance to malaria, as miR-451 affects parasite growth by modulating translational pathways in malaria infection [13]. In addition, studies have shown that the host miR-101b-3 can affect nematode parasitism through the regulation of superoxide dismutase 3 [14]. MiRNA in the host serum after a parasite infection can be used as a novel biomarker, such as miRNA in the blood as a test index for malaria infection [15], and the host serum miR-223 can be used as a marker of the degree of liver pathology of schistosomiasis infection [16]. 

There are some studies of miRNA transcriptome in pigeons, such as crop milk production [4] and pectoral muscle development [17]. However, there are no reports on miRNA transcriptome studies in pigeon’s resistance ability to *T. gallinae*. We have found that pigeons have different resistance abilities to *T. gallinae* in different ages and breeds, so the selection and breeding of *T. gallinae* resistance strains are important. In a previous report, we have studied the role of lncRNAs and mRNAs in resistance to parasitic infection in pigeons [18]. To improve our knowledge of host resistance to *T. gallinae*, we performed miRNA transcriptome sequencing analysis of oral mucosal tissues of pigeons of different resistances. The objectives of this research were to (1) evaluate miRNA expression profiles in the oral mucosa of pigeons; (2) identify differential miRNAs and their target genes; and (3) analyze the functional pathways of differential target genes. These would provide a basis for the further exploration of the mechanism of pigeon resistance to *T. gallinae*.

## 2. Results

### 2.1. Identification and Characterization of miRNAs

At 3 days post infection (dpi), the number of parasites in the oral cavity of the susceptible group was (3.19 ± 1.17) × 10^4^ but 0 in the tolerant group [18]. Via high-throughput sequencing, the transcriptome data of the control (C), susceptible (S) and tolerant (T) groups were obtained. The data were analyzed and filtered. A series of sequences that did not meet the requirements, such as containing the joint and being low quality, in the original sequence were filtered out, and the high-quality sequence obtained is shown in Table 1. After removing the joint, the lengths less than 18 bp and greater than 30 bp were removed. Finally, the clean reads of each sample were obtained. The data cleanliness rate was above 95% (Table 1). 

After the data quality control screening, the clean reads were compared with the reference genome of pigeons. The comparison rate was above 60%, of which the positive chain alignment rate was about 20%, and the negative chain alignment rate was about 50% (Table 2).

A total of 2429 miRNAs were obtained from all samples, including 1162 known miRNAs and 1267 newly predicted miRNAs (Appendix A). In the boxplot of overall distribution of miRNA expression and density distribution of miRNA expression, we could see that each sample has a good correlation, and samples in the same group have good correlations, as shown in Figure 1a,b. The TPM density distribution map showed that different sample curves had a high degree of coincidence (Figure 1c). The miRNA expression pattern of the samples was similar, and the sequencing results were reliable.

### 2.2. Differential Expression miRNAs

To explore the mechanism of resistance to *T. gallinae*, we focused on the comparison of the C and T groups. A total of 16 known miRNAs and 39 novel miRNAs were differentially expressed when comparing the C group to the T group. Among them, 19 miRNAs expression were down-regulated, and 36 miRNAs expressions were up-regulated. Novel_miR_741 was one of the down-regulated miRNAs in the T group compared to the C group (Figure 2a, Appendix A). 

We also compared the differential expression of the S group to the T group to explore the resistance mechanism. A total of 79 miRNAs were differentially expressed. There were 17 known miRNAs and 62 novel miRNAs. Among them, 45 miRNA expressions were down-regulated, and 34 miRNA expressions were up-regulated (Figure 2b, Appendix A). 

Further, we compared the differential expression of the C group to the S group. There were 77 miRNAs expressed differentially. A total of 48 miRNAs were novel miRNAs and 29 miRNAs were known miRNAs. A total of 61 miRNAs were up-regulated and 16 miRNAs were down-regulated (Figure 2c, Appendix A). 

The differentially expressed genes obtained from the three groups were statistically analyzed and a Wayne diagram was made. Among them, 16 miRNAs overlapped in the C vs. T group and S vs. T group, 21 miRNAs overlapped in the C vs. S group and C vs. T group, and 25 miRNAs overlapped in the C vs. S group and S vs. T group (Figure 2d).

### 2.3. miRNA Target Prediction and Analysis

A total of 2429 miRNAs were obtained as shown in Table 3, among which there were 1162 known miRNAs and 1267 newly predicted miRNAs. A total of 2121 miRNAs were predicted to be target genes, and a total of 22,054 target genes were finally obtained, including 9855 known miRNA target genes and 21,920 newly predicted miRNA target genes. The predicted target gene information of the three groups of differentially expressed miRNAs is shown in Appendix A. Moreover, *PRKCQ*, the target gene of novel_miR_741, related to the immune function, was up-regulated in the T group compared with the C group (Appendix A).

### 2.4. GO and KEGG Enrichment Analysis of Target Genes of Differentially Expressed miRNAs

To further understand the functions of differentially expressed miRNAs, miRNA target genes were enriched for analysis. For the C group and T group (Figure 3a), in the GO analysis, a total of 2554 genes were identified in the biological process, the cellular component, and the molecular function. The biological process category mainly contained the cellular, single-organism and metabolic processes. The cellular component category mainly contained the cell, organelle and membrane. The molecular function category mainly contained the binding, catalytic and transporter activity. For the susceptible and tolerant groups (Figure 3b), in the GO analysis, a total of 3105 genes were identified. For the control and susceptible groups (Figure 3c), in the GO analysis, a total of 3105 genes were identified.

The target genes of differential miRNAs were classified to identify pathways according to the KEGG functional annotation (Figure 3d–f). These genes were all annotated to the neuroactive ligand–receptor interaction, regulation of actin cytoskeleton, endocytosis, herpes simplex infection, tight junction, insulin signaling pathway and melanogenesis, and some of the pathways were associated with infection and immunity.

### 2.5. Novel_miR_741 Down-Regulate the Transcriptional Activity of PRKCQ

To detect whether novel_miR_741 negatively affected the transcriptional activity of *PRKCQ*, 293T cells were co-transfected with bisluciferase reporter plasmids and novel_miR_741 mimics. The activity of fireflies and sea kidney luciferase was determined 48 h after transfection. In cells transfected with mimics containing the *PRKCQ* 3′-UTR plasmid (WT), the luciferase activity was reduced, indicating that the mimics silenced the transcriptional activity of *PRKCQ* (Figure 4). In cells with plasmids mutated with *PRKCQ* 3′-UTR (MUT) sequences, the luciferase activity was unchanged (Figure 4). The results showed that the transcriptional activity of *PRKCQ* was down-regulated by novel_miR_741.

## 3. Discussion

*T. gallinae* has a serious impact on the meat pigeon industry, and it has been found that some pigeons are tolerant to *T. gallinae* infection [18]. The infection processes may alter various gene expression processes in cells; however, there is a lack of research on miRNA, which plays an important role in parasitic infection resistance [19]. In this study, the miRNA expression profile of the oral mucosa infected by *T. gallinae* of pigeons was designed to analyze the potential genes and related miRNAs. Previous studies in pigeon miRNAs have been focused on crop tissues [4], pectoral muscle and ovaries [20]. In the study of crop tissues, miR-20b-5p, miR-146b-5p, miR-21-5p and miR-26b-5p were found to regulate lactation [4]. Cli-miR-21-5p, cli-miR-148a-3p, cli-miR-10a-5p and cli-miR-26a-5p were reported to be associated with lactation by analyzing exosome miRNAs during the lactation period [21]. By studying the pectoral muscle miRNAs of pigeons at different age stages, 89 miRNAs and pathways of MAPK and TGF-β were found to be related [17]. MiR-205b was thought to play a key role in monochromatic light regulation of pigeon egg production [20]. 

Among the differentially expressed miRNAs, especially in the comparison of the C and T groups. The up-regulated miR-122 was the known miRNA with different species prefixes. MiR-122 was found to be abundantly expressed in mammalian livers and played an important role in hepatitis C viral replication and fatty acid and cholesterol metabolism [22,23]. MiR-122 could inhibit the expression of the gene *Sirt1* and regulate the LKB1/AMPK pathway, thereby promoting hepatic lipogenesis [24]. When mice were infected with Leishmania, parasite targeting was observed to lead to the down-regulation of mir-122 expression, and when mir-122 was injected in vitro, it could significantly reduce the hepatic parasite load in mice [25]. After coccidiosis infection in chickens, gga-miR-122-5p expression was up-regulated which was considered to be used as a diagnostic method of sub-clinical coccidiosis [26]. Cfa-miR-23b was a down-regulated, known miRNA in the analysis. MiR-23b was involved in regulating the normal physiological function, cell differentiation and cellular immunity [27]. These results indicated that miR-122 and miR-23b might play roles in regulating the endocrine system of pigeons during *T. gallinae* infection.

The GO and KEGG analyses were carried out on targeted genes of differentially expressed miRNAs, and some corresponding miRNAs were found to screen for immune-related genes. The analysis yielded 14 miRNAs, and all of them were novel-miRNA, of which 8 were down-regulated and 6 were up-regulated. The most obvious up-regulated miR-331 was confirmed to inhibit the proliferation and metastasis of cancer cells [28]. MiR-173 was thought to be associated with insect molting [29]. MiR-208 promoted cell proliferation in human esophageal phosphorus carcinoma by inhibiting the gene *SOX6* expression [30]. MiR-208 also inhibited apoptosis in mice with myocardial infarction [31]. MiR-936 suppressed cell invasion, proliferation of glioma, non-small cell lung cancer and laryngeal squamous cell carcinoma [32]. MiR-583 inhibited prostate cancer cell proliferation and invasion by targeting *JAK1* [33]. Based on the related studies of the above miRNAs, it could be speculated that these novel miRNAs might play a certain role in the resistance and immunity of trichomonosis in the process of *T. gallinae* infection.

The target gene of novel_miR_741, *PRKCQ*, also known as Protein kinase C (PKC), was confirmed to be involved in diverse cellular signaling pathways. *PRKCQ* is widely expressed in all kinds of cells, primarily in immune cells such as T cells and NK cells [34]. *PRKCQ* is crucial in the immune response due to it being required for maturation of Th17 cells, Ca^2+^ signaling, *NFAT* and *NFκB* activation [35,36,37,38]. Evidence supports that *PRKCQ* is essential in activating immune response for its role in activating T cell activity, so it is dispensable for immunity against viral and bacterial pathogens [39]. In our research, *PRKCQ* was negatively regulated by novel_miR_741, and the expression of *PRKCQ* was significantly higher in the T group than that in the C group, suggesting that the T group may activate a stronger immune response to eliminate *T. gallinae*. The individual could enhance immunity by reducing the expression of novel_miR_741 and improving the expression of *PRKCQ*.

The infection rates of trichomonosis were in the production range from 22% to 74%, so the disease has a great impact on the economy of the pigeon industry. Trichomoniasis is mainly treated with drugs such as metronidazole. Metronidazole is a carcinogen and inhibited from use in the pigeon industry. This experiment found a valid miRNA and its targeted genes, as novel_miR_741 and *PRKCQ*, related to the immune resistance of *T. gallinae* in pigeons, and these *T. gallinae* resistant genes could be used for molecular marker-assisted selection in breeding *T. gallinae* resistant lines in pigeon. Resistant lines would not be infected by *T. gallinae* easily, and thus, the economic benefit and food safety will be improved.

## 4. Materials and Methods

### 4.1. T. gallinae Inoculation and Examination

All 135 one-day-old White King pigeons were randomly divided into the control group (*N* = 35) and treatment group (*N* = 100), raised in two separate negative pressure isolators. The treatment group was nasally inoculated with 0.5 mL of 5 × 10^6^ parasites/mL of *T. gallinae* strain at 1, 2 and 3 days continuously. At 1, 2, 3, 4 and 7 days after the first inoculation, oral swab was collected from each squab and placed in 1.5 mL saline solution (pH 6.8) at room temperature. The number of parasites was counted with hemocytometer under microscope. The control group (C) was inoculated nasally with the same volume of parasite-free strain culture medium. The trial lasted for 15 days. At 3 days post infection (dpi), the pigeons in the infected group were divided into susceptible and tolerant groups based on the presence or absence of *T. gallinae* in the bird’s mouth. Four birds containing the greatest number of *T. gallinae* were selected as the susceptible group (S), and four birds with no parasites in their mouths were selected as the tolerant group (T). Three birds from the C group were selected. The selected squabs of C, S and T groups containing 3, 4 and 4 birds each were euthanized by cervical dislocation, and oral mucosal samples were collected. The rest of the squabs were raised to the 8th day for *T. gallinae* counting. 

### 4.2. RNA Isolation and Quality Assessment

Collected oral mucosal samples from C, S and T groups at 3 DPI were stored in liquid nitrogen. Total RNA was extracted using a Trizol reagent (Invitrogen, Carlsbad, CA, USA). The concentration and integrity of RNA were estimated using the NanoDrop 2000 (Thermo, Waltham, MA, USA) and Agilent 2100 Bioanalyzer (Agilent Technologies, Santa Clara, CA, USA), respectively.

### 4.3. RNA Sequencing and Quality Control

A 15% agarose gel was used to extract small RNAs (18–30 nt) from the total RNA. Ethanol precipitation and centrifugal enrichment of the small RNAs were carried out. Library building kits (VAHTS^TM^ Small RNA Library Prep Kit for Illumina^®^; Vazyme, Nanjin, China) were used to sequence libraries after they were established. Building libraries mainly included (1) reverse transcription of RNA; (2) PCR amplification and product purification; and (3) PAGE gel electrophoresis and gel recycling. Then, the library was qualified with Qsep-400. The established library was sequenced using Illumina NovaSeq 6000 platform (San Diego, CA, USA) and 50-bp single-end reads were obtained. The original image data file obtained using the Illumina platform sequencing was converted into the Raw Data by Base Calling, by removing the sample low-mass value sequence, removing unknown bases greater than 10% sequences, removing 3′ connector sequence and removing the sequences <18 nt and >30 nt to obtain clean reads.

### 4.4. miRNA Identification

The clean reads were aligned with the Silva database, GtRNAdb database, Rfam database and Repbase database using Bowtie [40], and miRNA unannotated reads were obtained after filtering. Columba_livia.Cliv_1.0 was used as a reference genome for sequence comparison to obtain mapped reads. The reads were aligned to the reference genome and mature sequences of known miRNAs including their upstream 2 nt and downstream 5 nt ranges in the miRBase (v22) database. The identified reads were considered as known miRNAs. Using the miRDeep2 (v2.0.0.5) software package, the possible precursor sequences obtained by aligning the position information on the genome with the reads were predicted as new miRNAs [41].

### 4.5. Differentially Expressed Analysis of miRNA

Differential expression analysis was performed using edgeR [42]. MiRNAs were considered to be differentially expressed when the |log_2_(Fold Change)| > 1.00 and the *p*-value < 0.05. Target gene prediction of the selected miRNAs was performed with miRanda (v3.3a) and TargetScan [43,44]. Using BLAST (v2.2.26), the predicted target gene sequences were aligned with the NR (ftp://ftp.ncbi.nih.gov/blast/db/FASTA/), Swiss-Prot (https://www.uniprot.org/), GO (https://geneontology.org/), KEGG (https://www.genome.jp/kegg/), KOG/COG (ftp://ftp.ncbi.nih.gov/pub/COG/KOG/kyva) and Pfam (v10.1) databases to obtain annotation information for target genes [45,46,47,48,49,50].

### 4.6. Luciferase Reporter Assays

A 3′-UTR fragment of the gene *PRKCQ* (XM_021281599) was cloned to the SacI/XhoII site of the GP-miRGLO vector to obtain a 3′-UTR luciferase reporter plasmid. At the same time, a 3′-UTR fragment mutation vector plasmid was constructed as a control group (synthesized by GenePharma, Shanghai, China). Luciferase assays were performed via cotransfection with 250 ng of 3′-UTR-luciferase reporter plasmid and miRNA mimics (10 nM) using GP-transfect-Mate (GenePharma, Shanghai, China). The 293T cells were assayed 48 h post-transfection for firefly and Renilla luciferase activities using the dual-luciferase assay. The Renilla luciferase values were then divided by the firefly luciferase activity values to normalize the difference in transfection efficiency. The experiments were performed in triplicate and repeated three times.

### 4.7. Data Accessibility

The sequencing data have been submitted to the SRA database (accession number: PRJNA909058). SRA records are accessible with the following link (https://www.ncbi.nlm.nih.gov/bioproject/PRJNA909058, accessed on 6 December 2022).

### 4.8. Statistical Analysis

Statistical analysis was conducted using SAS (v9.2). All values were presented as mean ± SEM, and *p* < 0.05 was considered as a significant difference, and *p* < 0.01 was considered as extremely significant difference.

## 5. Conclusions

Our study described miRNA expression profiles of oral tissues in pigeons from the C, S and T groups. The miRNA and target genes related to trichomonosis infected pigeons were analyzed. We demonstrated that novel_miR_741 had a targeting relationship with *PRKCQ* and could down-regulate the expression of the gene. The expression of *PRKCQ* was significantly higher in the T group than that in the C group, suggesting that the T group may activate a stronger immune response to eliminate *T. gallinae*. The above studies provide a theoretical basis for the regulatory mechanism of trichomonosis in pigeon infection.

## Figures and Tables

**Figure 1 ijms-24-16453-f001:**
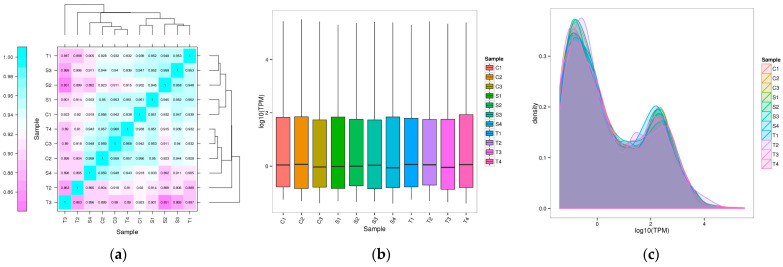
(**a**) Sample correlation diagram, where different colors represent different correlation values. The horizontal ordinate coordinate represents a different sample. (**b**) Overall distribution of miRNA expression in each sample Boxline Diagram. (**c**) The expression density distribution of miRNAs in each sample, the curves of different colors represent different samples, the horizontal coordinate of the points on the curve represents the logarithmic value of the corresponding sample TPM, and the ordinate coordinate of the point represents the probability density. C: control group; S: susceptible group; T: tolerant group.

**Figure 2 ijms-24-16453-f002:**
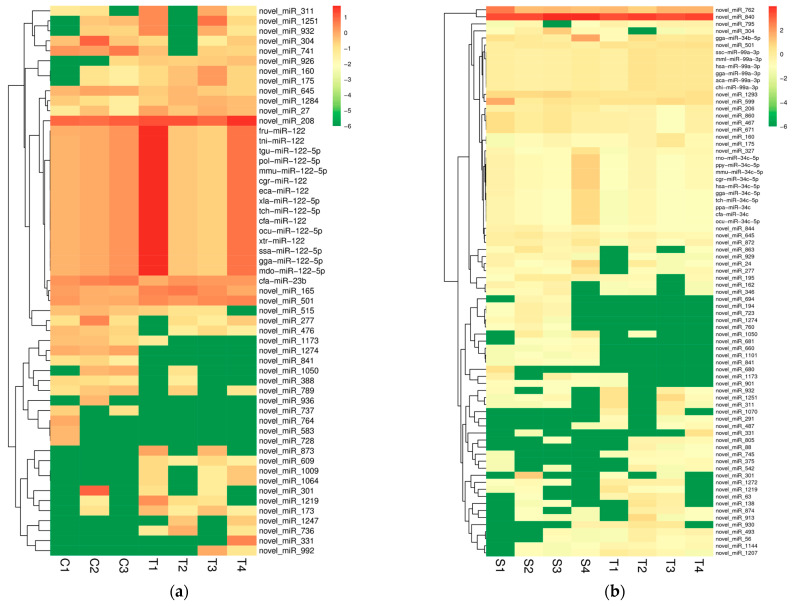
(**a**) Differentially expressed miRNA heatmaps between C group and T group. (**b**) Differentially expressed miRNA heatmap between S group and T group. (**c**) Differentially expressed miRNA heatmap between C group and S group. (**d**) Waven diagram of miRNAs expressed differently in C group, T group and S group. C: control group; S: susceptible group; T: tolerant group.

**Figure 3 ijms-24-16453-f003:**
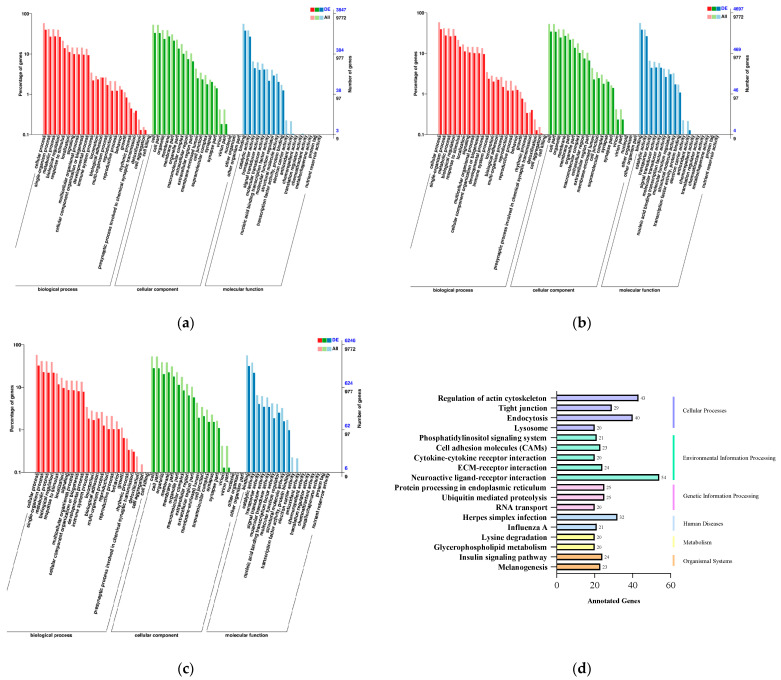
(**a**) GO enrichment analysis between C group and T group; (**b**) GO enrichment analysis between S group and T group; (**c**) GO enrichment analysis between C group and S group; (**d**) KEGG enrichment analysis between C group and T group; (**e**) KEGG enrichment analysis between S group and T group; (**f**) KEGG enrichment analysis between C group and S group. C: control group; S: susceptible group; T: tolerant group.

**Figure 4 ijms-24-16453-f004:**
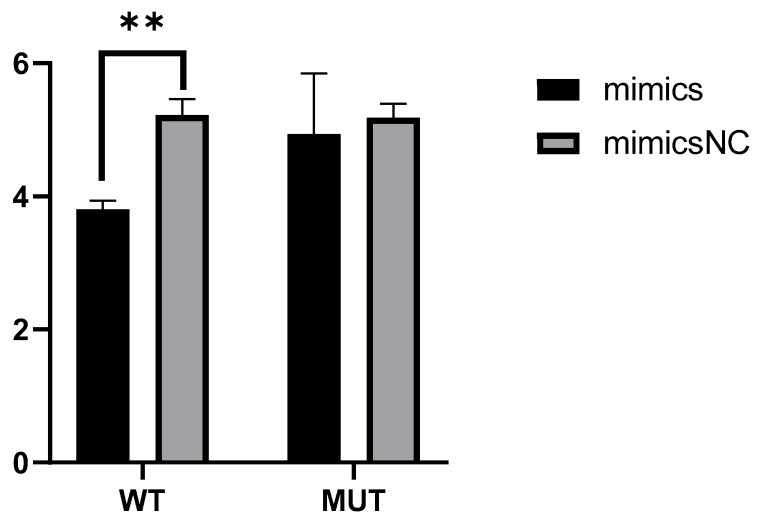
Diluciferase reports experimental results. WT: cells transfected with mimics containing the *PRKCQ* 3′-UTR plasmid; MUT: cells with plasmids mutated with *PRKCQ* 3′-UTR; ** *p* < 0.01.

**Table 1 ijms-24-16453-t001:** Basic data of miRNA sequencing.

Samples ^1^	Raw Reads	Low Quality	Containing ‘N’ Reads	Length < 18	Length > 30	Clean Reads	Q30 (%)
C1	26,719,180	0	0	140,044	0	26,579,136	95.47
C2	27,186,806	0	0	131,113	0	27,055,693	94.80
C3	32,297,099	0	0	201,598	0	32,095,501	95.84
S1	29,038,069	0	0	122,007	0	28,916,062	96.75
S2	33,717,404	0	0	250,965	0	33,466,439	97.46
S3	30,511,399	0	0	219,741	0	30,291,658	96.65
S4	30,764,981	0	0	49,885	0	30,715,096	96.33
T1	31,278,720	0	0	184,431	0	31,094,289	96.95
T2	33,971,942	0	0	237,057	0	33,734,885	97.19
T3	31,791,900	0	0	125,872	0	31,666,028	96.75
T4	34,630,509	0	0	106,722	0	34,523,787	96.78

^1^ C: control group; S: susceptible group; T: tolerant group.

**Table 2 ijms-24-16453-t002:** Mapping data of miRNA sequencing.

Samples ^1^	Total_Reads	Mapped_Reads	Mapped_Reads (+)	Mapped_Reads (−)
C1	22,132,444	15,780,016 (71.30%)	4,616,447 (20.86%)	11,163,569 (50.44%)
C2	24,296,565	17,895,382 (73.65%)	4,593,190 (18.90%)	13,302,192 (54.75%)
C3	29,065,079	21,596,946 (74.31%)	6,078,851 (20.91%)	15,518,095 (53.39%)
S1	24,687,783	18,275,595 (74.03%)	4,900,642 (19.85%)	13,374,953 (54.18%)
S2	27,049,727	19,922,935 (73.65%)	6,423,420 (23.75%)	13,499,515 (49.91%)
S3	25,404,509	18,372,520 (72.32%)	5,664,517 (22.30%)	12,708,003 (50.02%)
S4	29,284,210	21,069,624 (71.95%)	4,626,527 (15.80%)	16,443,097 (56.15%)
T1	22,943,862	16,663,501 (72.63%)	4,499,778 (19.61%)	12,163,723 (53.02%)
T2	26,130,297	16,922,718 (64.76%)	6,781,695 (25.95%)	10,141,023 (38.81%)
T3	26,485,112	19,276,440 (72.78%)	4,619,085 (17.44%)	14,657,355 (55.34%)
T4	30,523,003	21,635,872 (70.88%)	6,087,472 (19.94%)	15,548,400 (50.94%)

^1^ C: control group; S: susceptible group; T: tolerant group.

**Table 3 ijms-24-16453-t003:** Prediction statistics of miRNA target gene number.

Types	All_miRNA ^1^	miRNA_with_Target ^2^	Target_Gene ^3^
Known_miRNA	1162	998	9855
Novel_miRNA	1267	1123	21,920
Total	2429	2121	22,054

^1^ All_miRNA: total number of miRNAs; ^2^ miRNA_with_Target: predict the number of miRNAs of the target gene; ^3^ Target gene: the number of predicted target genes.

## Data Availability

The sequencing data of miRNAs have been submitted to the SRA database (accession number: PRJNA909058), and they are accessible with the following link https://www.ncbi.nlm.nih.gov/bioproject/PRJNA909058, accessed on 6 December 2022. Other data used and analyzed during the study are accessible in the Appendix A and available from the corresponding author.

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
