# Peer review of "Identification of miRNA Associated with Trichomonas gallinae Resistance in Pigeon (Columba livia)"

_ijms, 2023, doi:10.3390/ijms242216453_

Round 1

Reviewer 1 Report

Comments and Suggestions for Authors

GENERAL

The authors investigated miRNA associated with Trichomonas gallinae in pigeons by classifying pigeons challenged with T. gallinae into Susceptible and Tolerant groups based on whether this pathogen was present in the mouths of the pigeons on day 3 after exposure. A Control group was used. The authors identified known and unknown miRNAs. The influence of novel_miR_741 on PRKCQ was studied.

The abstract should be submitted for English editing to improve the use of punctuation and increase readability.

Methodology  - How did was T. gallinae detected in the mouths of the treatment group pigeons?

SPECIFIC

The text on the Axes of Figure 3 is too small to be read

Reference 18 - only 2 authors were listed

Comments on the Quality of English Language

The language can be improved. Attention should be paid to punctuation.

Reviewer 2 Report

Comments and Suggestions for Authors

The authors addressed an extremely interesting, current topic, which discusses the sector of breeding pigeons for meat, its problems, and a possible solution.

Were the pigeons in the study artificially infected? To what extent the experiment can be extrapolated to naturally infected pigeons? Through the results of this study, can the economic impact be estimated in the sector of breeding pigeons for meat from the authors' point of view?

This aspect should be highlighted.

From the analysis of the manuscript, I recommend publication after minor revision.
